# Performance of the Safer Nursing Care Tool to measure nurse staffing requirements in acute hospitals: a multicentre observational study

Peter Griffiths [ORCID],[1,2] Christina Saville [ORCID],[1] Jane Ball,[1] David Culliford,[1,2] Natalie Pattison,[3,4] Thomas Monks[2,5]

[1]School of Health Sciences, University of Southampton, Southampton, UK
[2]National Institute for Health Research Collaboration for Leadership in Applied Health Research and Care Wessex, University of Southampton, Southampton, UK
[3]Department of Clinical Services, Royal Marsden NHS Foundation Trust, London, UK
[4]School of Health and Social Work, University of Hertfordshire, Hatfield, UK
[5]University of Exeter Medical School, University of Exeter, Exeter, UK

**Correspondence to**
Dr Christina Saville;
C.E.Saville@soton.ac.uk

## ABSTRACT

**Objectives** The best way to determine nurse staffing requirements on hospital wards is unclear. This study explores the precision of estimates of nurse staffing requirements made using the Safer Nursing Care Tool (SNCT) patient classification system for different sample sizes and investigates whether recommended staff levels correspond with professional judgements of adequate staffing.

**Design** Observational study linking datasets of staffing requirements (estimated using a tool) to professional judgements of adequate staffing. Multilevel logistic regression modelling.

**Setting** 81 medical/surgical units in four acute care hospitals.

**Participants** 22 364 unit days where staffing levels and SNCT ratings were linked to nurse reports of "enough staff for quality".

**Primary outcome measures** SNCT-estimated staffing requirements and nurses' assessments of staffing adequacy.

**Results** The recommended minimum sample of 20 days allowed the required number to employ (the establishment) to be estimated with a mean precision (defined as half the width of the CI as a percentage of the mean) of 4.1%. For most units, much larger samples were required to estimate establishments within ±1 whole time equivalent staff member. When staffing was lower than that required according to the SNCT, for each hour per patient day of registered nurse staffing below the required staffing level, the odds of nurses reporting that there were enough staff to provide quality care were reduced by 11%. Correspondingly, the odds of nurses reporting that necessary nursing care was left undone were increased by 14%. No threshold indicating an optimal staffing level was observed. Surgical specialty, patient turnover and more single rooms were associated with lower odds of staffing adequacy.

**Conclusions** The SNCT can provide reliable estimates of the number of nurses to employ on a unit, but larger samples than the recommended minimum are usually required. The SNCT provides a measure of nursing workload that correlates with professional judgements, but the recommended staffing levels may not be optimal. Some important sources of systematic variations in staffing requirements for some units are not accounted

### Strengths and limitations of this study

- ► In contrast to most studies of nurse staffing tools, this was a large-scale study conducted in 81 units in four hospitals over a year.
- ► This is the first study to provide an independent evaluation of this tool, which is used in most hospitals in England.
- ► The observational study measured associations between staffing shortfalls measured using the SNCT and subjective professional assessments of staffing adequacy.
- ► The study did not explore the impact on objective care outcomes.

for. SNCT measurements are a potentially useful adjunct to professional judgement but cannot replace it.

**Trial registration number** ISRCTN12307968.

## INTRODUCTION

In acute care hospitals, the ability to determine the 'right' number of nursing staff to employ and to deploy on any given shift is an imperative, as nurse staffing levels influence both efficiency and quality of care delivery. On the one hand, professional nurses and nursing support staff form the largest group of staff and the largest variable costs faced by hospitals. Nursing budgets are thus frequently targeted in the drive for cost savings.[1] On the other hand, inadequate nurse staffing is linked to deficits in the quality and safety of care.[2] However, despite the existence of many tools to determine staffing requirements and an extensive literature, evidence about the ability of any tool to reliably and accurately estimate staffing requirements is extremely limited.[3 4] In this paper, we consider the 'Safer Nursing Care Tool' (SNCT), which is used in the majority of acute hospitals in the UK's National Health Service (NHS)[5] and endorsed by the National Institute for Health

and Care Excellence (NICE),[6] the body that produces evidence-based guidelines for the NHS. We explore the reliability and precision of the estimates of required staffing establishments (ie, the number of nurses to employ for a hospital unit) made using the tool and the extent to which estimated staffing requirements correspond with professional judgement of sufficient staffing. Despite the tool's widespread use and the importance of these considerations, these factors have not been previously studied.

Associations between higher registered nurse staffing levels in hospitals and improved care quality have been demonstrated in many studies.[2 7–9] Outcomes include lower risks of in-hospital mortality,[10] shorter lengths of stay[11] and fewer omissions of necessary care.[12] Such findings have underpinned policies to make minimum nurse staffing levels mandatory in some jurisdictions, for example, in California in the USA, some Australian states and more recently in Germany.[13] Yet studies showing associations between nurse staffing levels and outcomes rarely provide a clear indication of how many staff are needed for different patients, despite evidence that needs can vary considerably, and few studies have explored tipping points in relationships, which could be one indication of an optimal staffing level.[2 14 15]

Consequently, tools and systems to guide decisions about the number of nursing staff to employ or to deploy on any given shift are still widely used either in conjunction with or as an alternative to mandatory minimums. At the heart of most tools is some form of patient-level assessment, which is translated into a measure of required nursing time.[3 4] Many such tools exist, although they are largely unsupported by robust evidence. Studies used to support the validity of tools to determine staffing requirements simply tend to show that staff demand estimated using a given tool correlates with some other measure of demand. In the absence of a gold standard, and without addressing whether the staffing according to the predicted level is sufficient to deliver the required care, such evidence is significantly limited. Different tools, while providing results that are highly correlated, can and do give dramatically different estimates of the staffing required by the same group of patients.[4] For example, applying a new system to estimate the staffing required for low acuity wards resulted in an estimate that was double that derived from the existing system.[16]

Furthermore, although a key driver for choosing to use a tool is the assertion that variable patient need cannot be efficiently met by fixed staffing levels,[17] little consideration has been given to the impact of variation on the resulting estimates of average staffing requirements. Inter-rater reliability and agreement is often reported, the precision with which the unit staffing requirement as a whole is estimated, either on a given day or over time, is not.[4]

The SNCT[18] is reported to be used in 80% of National Health Service acute hospitals in England.[5] The tool was originally designed to determine the required number of staff to employ (the establishment) for each unit to ensure that there are sufficient staff to fill daily rosters to meet average need, but it is increasingly used to monitor and determine daily demand for staff. It is not however used for billing purposes since in England billing is based on activity and does not explicitly account for nursing staff. The SNCT is an example of a patient classification system.[4] At least once per day, patients occupying beds on the ward are classified into one of five groups, based on their acuity and dependency on nursing care, with each group having an associated weighting (described as a 'multiplier') indicating the number of nursing staff required.[18] At the time of writing, the most recently published multipliers for general adult inpatient units were based on observations of 40 000 patient care episodes.[6] The multipliers represent the average of staff time to provide all direct patient care and ancillary work for patients in each group with allowances made for annual leave, study time and sickness absence when determining the number of nurses to employ.[18]

The SNCT has been shown to correlate strongly with an alternative classification system and high inter-rater agreement is reported.[19 20] However, while the tool's handbook recommends a sample of at least 20 days to establish a reliable baseline for setting establishments, we could find no evidence of the precision of the resulting estimates. In our review of literature, we found no direct evidence that using this or any other tool improved the quality of care.[4] Therefore, we used professional judgement as the 'gold standard', as we found no evidence that any tool provides a more accurate measure of the staffing required.

This observational study aims to provide evidence about the reliability and validity of the SNCT by addressing the precision of the estimated establishment and the extent to which staffing shortfalls relative to the level implied by the tool are associated with nurses' judgements that staffing levels are sufficient to deliver all necessary care with acceptable quality. Because factors such as patient turnover, specialty and layout are not directly considered in patient classifications yet may influence staffing requirements,[21–23] we also examine the extent to which staffing levels determined using the SNCT are sufficient to accommodate variation in demand associated with these factors by determining whether there is an independent association between these factors and judgements of staffing adequacy when considering the effect of shortfalls from the SNCT recommended staffing level.

## METHODS AND MATERIALS

This paper draws on research and data as described in detail previously in the NIHR *Journals Library Health Services and Delivery Journal*.[24] We used routinely collected data and nurse reports over 1 year (2017) from 81 acute medical/surgical units (2178 beds) in four NHS hospital trusts (hereafter referred to as hospitals for brevity) in England. For each unit and for each day, we identified staffing measurements: the staffing level deployed (from

the electronic roster), the staffing level required (based on patient classifications using the SNCT) and nurses' professional judgement of the completeness of care and adequacy of staffing to deliver quality (through a microsurvey embedded in the daily assessments). SNCT and staffing adequacy assessments were provided by the nurse in charge of the shift, hereafter referred to as 'shift leader' for brevity.

## Setting and inclusion

The study sites were one university teaching hospital, two general hospitals and a specialist cancer hospital (two sites) based in London, South East and South West England. The hospitals serve diverse populations including rural areas, deprived inner city populations and specialist national referrals. All hospitals undertook reviews of nurse staffing establishments at least twice a year. Two had been using the SNCT as part of this process for some time, while two adopted it shortly before the study commenced.

We included general medical and surgical units that provided 24-hour inpatient care. Services out of scope of the SNCT (eg, paediatrics, intensive care, maternity, neonatal and palliative care) and any others with highly atypical staffing requirements (eg, bone marrow transplant and isolation units), as determined by a local co-investigator, were excluded. Our unit sample represented 74% of all beds across the four hospitals.

## Data sources and measures

Over the course of 1 year at least twice per day, shift leaders recorded the number of patients in each SNCT category and made judgements about staffing adequacy (see below) in electronic systems. Local leads trained potential shift leaders on participating units in the use of the SNCT and completion of the staffing adequacy questions. Supporting information and brief guidance was provided on laminated sheets kept near the unit computers where data were entered. Other data for the study were routinely collected for administrative purposes (roster, patient admissions or discharges). Each hospital supplied a profile for each unit with main specialty and layout including the number of beds/single rooms.

## Study variables

We used the most up-to-date SNCT multipliers available at the start of the study.[18] We took the reported counts of patients in each category and calculated the weighted average multiplier per unit and day. We multiplied this by the patient count derived from the patient administration system, in case any patients were omitted from shift leader reports. This figure provides an estimate of the required unit establishment (number of staff to employ). We used morning assessments (substituting later assessments from the same day if missing) and patient count at 07:00 for our main analysis. The SNCT calculation gives the number of staff to employ (staffing establishment), including an uplift for staff leave and an allowance for sickness, so we

converted this into the implied daily hours of staff time available. For this, we used a 37.5-hour working week for one whole time equivalent and removed the 22% 'uplift', which is added to the SNCT establishment to account for holidays, study and sick leave. We assumed that this uplift used in the tool was enough for the staff employed to be able to cover all long-term absences.

For each unit, we used the average observed skill mix on that unit as a proxy for the planned skill mix of registered nurses and nursing assistants. The SNCT does not directly account for patients identified as requiring one-to-one supervision, often referred to as 'specialing'[25] even though the implied staffing requirement is very high. Therefore, where we wanted to identify the staffing requirements for any particular day, we identified the number of such patients from records and added the required hours to our estimated staffing requirement. However, because such enhanced care would form part of the care observed to determine the SNCT multipliers and thus be included in the average, we made no additional allowance when estimating establishments to be employed.

From the electronic roster, we identified hours worked by registered nurses and nursing assistants each day (from 07:00 to 07:00) and divided these by the number of patient days (patient hours/24) to calculate hours per patient day (HPPD) for each unit for each day. We calculated a measure of staff shortfall by subtracting the required hours (according to the SNCT plus specialing requirements) from the hours actually deployed on that day. If more staff than the estimated requirement were deployed, the shortfall was negative. We also calculated daily patient turnover per staff member (the numbers of patients entering and leaving units divided by the total staff hours).

The outcomes measured were a number of variables reflecting the adequacy of nurse staffing, as reported by the nurse in charge of the shift ('shift leader'). The shift leader responded to three brief items every time they provided SNCT ratings, directly inputting responses into the same system as used for SNCT (see box 1). We chose to use three items for pragmatic reasons, since we judged that more would have been too much of an administrative burden and might have resulted in poorer data quality. Two items, based on the widely used RN4CAST/International Hospital Outcomes surveys of nurse staffing and quality, asked whether there were enough staff for quality

> **Box 1  Staffing adequacy questions**
>
> **Questions**
> ► Were there enough nursing staff to provide quality care on the last shift?
> ► Was necessary nursing care left undone (missed) on the last shift because there were too few nursing staff?
> ► Were staff breaks missed on the last shift because there were too few nursing staff?

and whether any necessary care was left undone.[12 26] We also asked about staff missing breaks, as nurses may miss breaks to complete care activities, creating additional staff time that avoids adverse effects of staffing shortfalls.[27] These questions constituted the microsurvey.

## Data cleaning and analysis

Data cleaning, processing and statistical analyses were carried out in R statistical software V.3.5.0.[28] We identified and removed extreme values of staffing shortfall, where values lay outside the mean ±3 SD (approximately 1.5% of cases). This removed atypical periods if the unit was not functioning as normal, for example, over the Christmas period, or where there is an extreme error in the recorded SNCT ratings. Where there were major changes such as unit moves, changes to the patient population or bed numbers, data for that unit were split and treated as separate units. We found some evidence of consistent reverse coding of data inputs (0/1 for yes/no) for some staffing adequacy questions in several units of one hospital. This appeared to result from erroneous staff training. Because it was discovered partway through the study, we developed logical rules to identify units where this occurred and recode data, considering the implications of this through sensitivity analyses where we excluded the hospital entirely.

To understand the accuracy of the estimated establishment, we first considered the minimum recommended sample size of the SNCT data collection, which is 20 days twice a year. We used 1000 bootstrap samples of 20 days' data to estimate a mean establishment with a 95% CI for the establishment on each unit. We repeated this with bootstrap samples of increasing numbers of days to assess the accuracy of larger samples. For each unit, we calculated both the precision of the estimate as half the width of the CI expressed as a % of the mean, and the absolute value of the CI in whole time equivalent (WTE) staff members. We determined the number of units where the CI of the establishment was 2 WTE or less (ie, no more than ±1 WTE difference from the mean) or 1 WTE or less (mean ±0.5).

We modelled the relationship between staffing deficits (in HPPD) and nurse-reported measures of staffing adequacy. For this we used the first available SNCT rating per day (morning or later time if missing) for the proportions of patients in each level, the 7am patient count, actual staffing and patient hours from 7am-7am, and the staffing adequacy recorded in the morning of the next day. We fitted multilevel logistic regression models for binary outcomes using the glmer (generalised linear mixed effects regression) function from the lme4 package[29] in R. Staffing was nested in unit which was nested in hospital. All models included control for day of the week, proportion of single rooms, turnover and unit specialty (surgical vs medical or mixed). We considered the association of staffing adequacy outcomes with deviation of both registered nurse staffing and nursing assistant staffing from their estimated requirements. We also fitted models using the deviation in total hours and skill mix (registered nurse proportion).

After modelling the linear and main effects, we introduced quadratic terms for the staffing level variables to assess non-linear relationships and we investigated whether staffing variables interacted with other variables. We compared the fit of models using the Akaike information criteria (AIC) and Bayesian information criteria (BIC), preferring models with lower values, indicating better fit/more parsimonious models.[30]

## Patient and public involvement

When developing our research proposal, we discussed appropriate patient/public involvement with Claire Ballinger (Patient and Public Involvement lead for the NIHR Collaboration for Leadership in Applied Health Research and Care [CLAHRC] Wessex) and Anya de Iongh (Patient and Public Involvement champion for the 'fundamental care in hospital' theme of the CLAHRC). Following their guidance, we sought no further direct patient/public involvement in prioritising or shaping the questions as these arose from the brief and need for technical assessment of the tool, instead we focused on considering how the public could be involved in the proposed research, its governance and dissemination.

Based on this advice, we sought a lay member of our steering group with specific interest and expertise. To this end, we worked with Stephen Habgood, a lay member of the NICE safe staffing advisory committee for the development of guidelines in mental health, who agreed to participate in the project steering group. Stephen also has experience of staffing methodologies used in other sectors from his past work as a prison governor and is currently chair of a mental health charity. Additionally, and guided by the advice from our patient and public involvement experts, we also considered ward-based staff nurses as the potential end users of our research in a way that is analogous to a patient receiving a treatment that might be recommended by an expert. Based on this, we have used multiple channels to connect with ward-based staff including extensive use of social media and discussions with individual staff nurses at consultation events.

## Ethical approval and registration

The study was prospectively registered.[31] This study did not require NHS Research Ethics Committee approval because no data were collected directly from patients, and all patient data were pseudoanonymised at source with no sensitive patient data transferred.

## RESULTS

We had useable SNCT ratings on 96% of occasions and responses to staffing adequacy questions on 85% or more of possible occasions. After data cleaning and linkage, we had 22 271, 22 294 and 22 364 unit days where we could assess the association between staffing shortfalls and

**Table 1** Mean and range of units' average daily staffing levels, skill mix and SNCT estimated staffing requirements

| Hospital | Total hours per patient day | | | Skill mix (% registered nurses) | | | Estimated staffing requirement | | |
|---|---|---|---|---|---|---|---|---|---|
| | Mean | Min | Max | Mean | Min | Max | Mean | Min | Max |
| A | 7 | 5.4 | 10.4 | 51 | 42 | 70 | 7.4 | 5.9 | 10.2 |
| B | 6.8 | 5.0 | 8.9 | 56 | 39 | 79 | 7.3 | 6.0 | 9.4 |
| C | 10.5 | 7.5 | 14.2 | 75 | 70 | 78 | 7 | 6.3 | 7.4 |
| D | 6.5 | 5.2 | 8.4 | 49 | 40 | 63 | 7 | 6.5 | 7.6 |
| All | 7.3 | 5.0 | 8.4 | 56 | 39 | 79 | 7.2 | 5.9 | 10.2 |

SNCT, Safer Nursing Care Tool.

reports of staff breaks missed, nursing care left undone and enough staff for quality, respectively.

Average unit staffing levels and skill mix varied considerably between hospitals and between units within hospitals (table 1). At a hospital level, average estimated staffing requirements of units corresponded closely with the observed staffing levels in 3 of 4 hospitals although all were somewhat understaffed relative to the estimated requirement (8% or less). Larger differences between actual staffing and SNCT estimates occurred on smaller, generally specialist, units with more single rooms (where apparent overstaffing occurred), and some larger medical units (where extremes of apparent understaffing occurred). In hospital C, a specialist hospital with many small units, average unit staffing was 50% higher than the SNCT estimated requirement.

Across all units, using the recommended minimum of 20 days' data, the average precision was 4.1% but varied by unit (range 0.6%–13.5%). In absolute terms, the average width of the 95% CIs for the establishment was 2.9 whole time equivalent staff (ie, approximately mean ±1.5 WTE). The CI width was ≤2 WTE in 27/86 units and ≤1 WTE in only 3/86.

As the number of days sampled increases from 20, there was a marked increase in precision (figure 1), with most units (56/86) yielding a CI width of ≤2 WTE from a sample of 40 days. The benefits of increased sample sizes diminishes with larger samples however, and even with samples of 180 days only 53/77 units gave a CI width that was ≤1 WTE wide (table 2).

Across all units, a mean of 78% of shifts were assessed by the nurse in charge as having enough staff to deliver quality care (unit range 24%–100%). Necessary nursing care was reported left undone because of too few staff on 5% of shifts (range 0%–25%), and breaks were reported missed on 5% of shifts (range 0%–29%).

Shortfalls in staffing levels relative to the requirement for that day, estimated using the SNCT, were associated with nurses' perceptions of staffing adequacy (table 3). In the multivariable models, for each registered nurse hour shortfall, the adjusted odds of the shift leader reporting that there were enough staff for quality were 11% lower, the odds of reporting nursing care left undone were increased by 14% and the odds of staff missing breaks were increased by 12%. Findings are similar for shortfalls of nursing assistants.

Factors other than shortfalls relative to the SNCT estimated requirement were also associated with perceptions of staffing adequacy. Nurses on surgical units were less likely to perceive adequate staffing compared with nurses on other (medical or mixed) units with lower odds of reporting enough staff for quality and higher odds or reporting care left undone or missed breaks. For example, the odds of nurses reporting that there were enough staff for quality were 46% lower on surgical units. Although relationships were not significant and CIs were wide, the odds of reporting enough staff for quality were substantially lower on units with a higher proportion of single rooms. Similarly, odds of reporting care left undone and missed breaks were substantially increased on units with a higher proportion of single rooms and units with higher turnover, although again CIs were wide and relationships were not statistically significant. Nurses were more likely to report that there were enough staff for quality and less likely to report missed care and missed breaks on Saturday compared with Monday, although there was no consistent pattern that suggested weekends differed from weekdays overall.

We tested for non-linear relationships with registered nurse and nursing assistant shortfalls, as would be expected if the SNCT provides a threshold for adequate staffing. We estimated models using effects that were significant in the main effects models, adding a variable for staffing shortfall squared for each staff group. For nursing care left undone, the non-linear term for registered nurse staffing was significant, but there was no clear indication that the model was preferable (AIC: Δ−2, BIC: Δ+14 compared to model with significant variables only) and the overall relationship was not changed substantially. No indication of a threshold where benefit/harm starts could be observed (see figure 2). For other outcomes, non-linear terms were not significant and associated with increased AIC and BIC (online supplementary table 1).

We estimated models that included all statistically significant variables and interactions between staffing shortfall and these variables. There were no significant interaction effects between registered nurse shortfall and nursing assistant shortfall and both AIC and BIC increased for

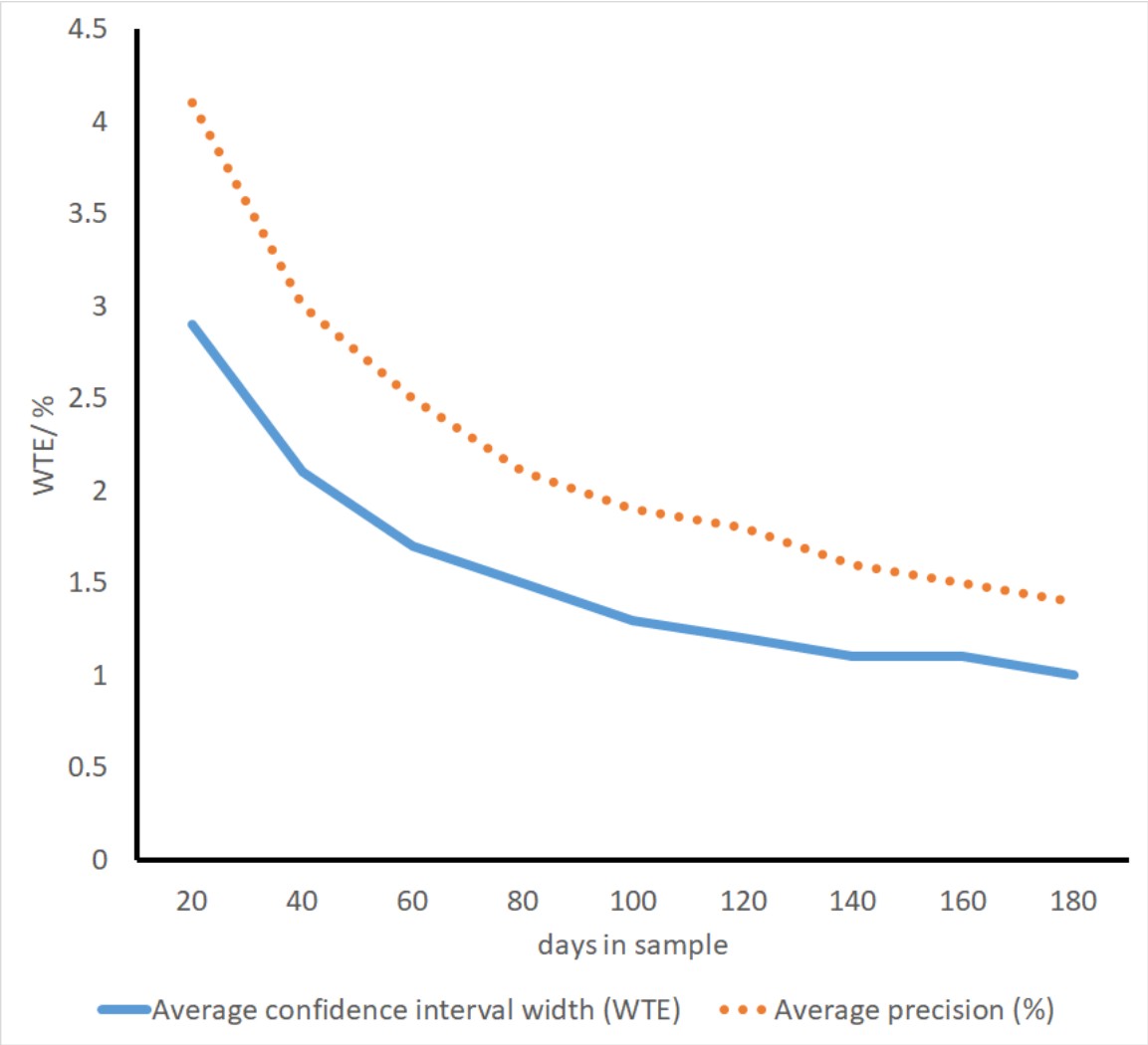

**Figure 1** Mean precision and CI width of staffing establishment estimates with different sample sizes. WTE, whole time equivalent.

| Table 2 | Average widths of 95% CIs for the mean using different sample sizes to estimate establishment | | | | |
|---|---|---|---|---|---|
| Sample size taken for the estimate | Average CI width (WTE) | Average precision (%) | Number units with CI width 1 WTE or less | Number units with CI width 2 WTE or less | Number of units* |
| 20 | 2.9 | 4.1 | 3 | 27 | 86 |
| 40 | 2.1 | 3.0 | 7 | 56 | 86 |
| 60 | 1.7 | 2.5 | 10 | 64 | 86 |
| 80 | 1.5 | 2.1 | 20 | 72 | 86 |
| 100 | 1.3 | 1.9 | 31 | 74 | 82 |
| 120 | 1.2 | 1.8 | 39 | 74 | 81 |
| 140 | 1.1 | 1.6 | 44 | 74 | 81 |
| 160 | 1.1 | 1.5 | 50 | 75 | 80 |
| 180 | 1.0 | 1.4 | 53 | 73 | 77 |

*Because units with establishment and/or specialty changes were treated as separate units for analysis, the total exceeds the number of units participating in the study. As the available data for some units was less than the sample required for the estimate, the number of units for larger samples is reduced.
WTE, whole time equivalent.

**Table 3** Association between staffing shortfall and nurse perceptions of staffing adequacy: univariable and multivariable models

| Variable | Enough staff for quality | | | | Nursing care left undone | | | | Staff breaks missed | | | |
|---|---|---|---|---|---|---|---|---|---|---|---|---|
| | OR* | Adjusted OR | 95% CI | P value | OR* | Adjusted OR | 95% CI | P value | OR* | Adjusted OR | 95% CI | P value |
| Registered nurse shortfall (HPPD) | 0.94 | 0.89 | (0.87 to 0.92) | 0.000 | 1.09 | 1.14 | (1.08 to 1.20) | 0.000 | 1.08 | 1.12 | (1.06 to 1.18) | 0.000 |
| Nursing assistant shortfall (HPPD) | 0.90 | 0.86 | (0.83 to 0.89) | 0.000 | 1.08 | 1.14 | (1.07 to 1.20) | 0.000 | 1.06 | 1.11 | (1.05 to 1.17) | 0.000 |
| Turnover (per nursing hour) | 0.35 | 0.91 | (0.30 to 2.75) | 0.863 | 7.16 | 3.36 | (0.60 to 18.72) | 0.167 | 3.85 | 4.94 | (0.95 to 25.78) | 0.058 |
| **Unit type** | | | | | | | | | | | | |
| Medical or mixed (ref) | 1.00 | 1.00 | | | 1.00 | 1.00 | | | 1.00 | 1.00 | | |
| Surgical | 0.57 | 0.54 | (0.31 to 0.92) | 0.023 | 2.00 | 2.13 | (1.19 to 3.82) | 0.011 | 2.13 | 2.15 | (1.23 to 3.78) | 0.008 |
| Proportion single rooms | 0.89 | 0.54 | (0.18 to 1.66) | 0.283 | 1.58 | 3.01 | (0.93 to 9.71) | 0.065 | 1.05 | 2.06 | (0.65 to 6.56) | 0.221 |
| **Day of week** | | | | | | | | | | | | |
| Monday (ref) | 1.00 | 1.00 | | | 1.00 | 1.00 | | | 1.00 | 1.00 | | |
| Tuesday | 1.11 | 1.12 | (0.99 to 1.26) | 0.079 | 0.86 | 0.86 | (0.69 to 1.06) | 0.160 | 0.94 | 0.71 | (0.58 to 0.88) | 0.001 |
| Wednesday | 1.28 | 1.28 | (1.13 to 1.45) | 0.000 | 0.95 | 0.96 | (0.78 to 1.18) | 0.684 | 1.01 | 0.61 | (0.49 to 0.76) | 0.000 |
| Thursday | 1.09 | 1.08 | (0.96 to 1.23) | 0.200 | 0.90 | 0.91 | (0.73 to 1.13) | 0.383 | 0.96 | 0.81 | (0.66 to 1.00) | 0.045 |
| Friday | 1.03 | 1.03 | (0.91 to 1.17) | 0.610 | 0.93 | 0.93 | (0.74 to 1.15) | 0.488 | 1.03 | 0.79 | (0.64 to 0.97) | 0.028 |
| Saturday | 1.28 | 1.29 | (1.14 to 1.47) | 0.000 | 0.74 | 0.75 | (0.60 to 0.95) | 0.000 | 0.86 | 0.50 | (0.40 to 0.64) | 0.000 |
| Sunday | 1.02 | 1.02 | (0.90 to 1.15) | 0.811 | 1.08 | 1.11 | (0.90 to 1.37) | 0.310 | 0.84 | 0.82 | (0.66 to 1.01) | 0.058 |
| Variance partition coefficient for units** | | 0.22 | | | | 0.22 | | | | 0.23 | | |
| Variance partition coefficient for hospitals** | | 0.12 | | | | 0.17 | | | | 0.11 | | |
| Akaike information criterion | | 20697 | | | | 8377 | | | | 8094 | | |
| Bayesian information criterion | | 20809 | | | | 8490 | | | | 8206 | | |

Ref: reference category for categorical variables.

*OR derived from entering this variable only into the multilevel model. **Calculated as between-group residual variance divided by total variance using the latent variable approach.[46]

HPPD, hours per patient day.

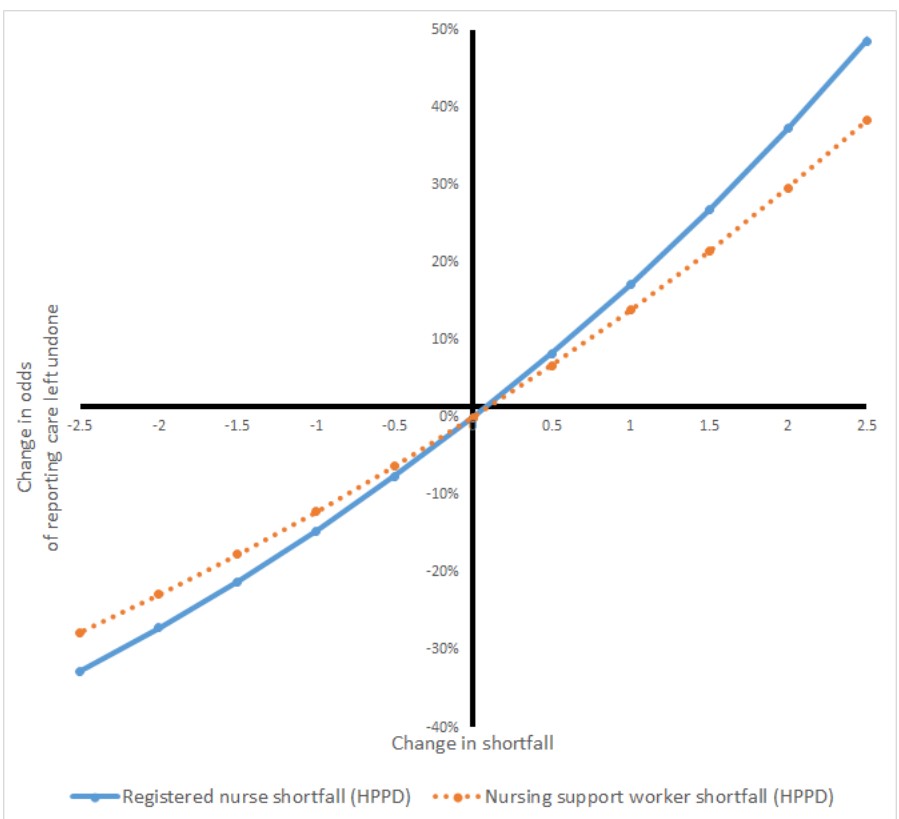

**Figure 2** Change in odds of reporting care left undone with change in staffing shortfalls estimated from model with non-linear staffing effects. HPPD, hours per patient day.

these models (online supplementary table 2), so the simpler models were preferred.

Models using overall care hour shortfall (registered nurse and assistants) per patient day gave similar coefficients, with each care hour per patient day of shortfall associated with a 12% reduction in the odds of reporting enough staff for quality and no significant associations with skill mix (online supplementary table 3). Because of the coding errors noted from units in one of the hospitals, we repeated the main models omitting data from this hospital. Results were largely unchanged with no effect on substantive conclusions (see online supplementary table 4 for an example).

## DISCUSSION
This study reports the first independent assessment of the SNCT, which is widely used to determine staffing levels in English hospitals. Using the recommended minimum 20-day sample, estimates for the number of nurses that should be employed on a ward had an average precision of 4.1%, but wide CIs for the absolute staff numbers needed. A sample of 40 days gave an estimate within ±1 staff members for the majority of wards, but much larger samples (140 days or more) are required to estimate the staff required with a CI width no more than one staff member wide in the majority of wards. When staffing shortfalls were high, relative to the required level estimated using the SNCT for that day, staff were less likely

to report that they had enough staff for quality and more likely to report that necessary nursing care was omitted and staff breaks missed. These relationships appeared to be linear, with no threshold when staffing reached the SNCT recommended level. Other factors, not included in the patient classifications used by the SNCT, including unit specialty and day of the week are also associated with whether a given staffing level is deemed to be sufficient by nurses working on the unit.

The original purpose of the SNCT was to ensure that units employed sufficient nurses to be able to provide the care hours required by patients. Existing reports attest to the inter-rater reliability of the tool,[19 20] but the recommendation that this staffing establishment is estimated using at least 20 days of SNCT data recognises that daily demand is variable and estimates based on small samples may be imprecise. The average level of precision achieved from 20 days of observations in our study appears, superficially, to be acceptable. However, this masks considerable variation in precision between units and large absolute differences in terms of numbers of staff. Using a conventional (if technically slightly inaccurate) interpretation of the CI, this means that for many units, estimates could vary from the true staffing requirement by more than two whole staff members. The absolute importance of such differences may vary by unit, but the potential significance of such inaccuracy is great.

Small increases in the number of days used to estimate establishments yield substantial improvements in precision, although there are rapidly diminishing returns from samples of more than 40 days. As more and more hospitals are gathering SNCT ratings on a daily basis, it may be that these data could be drawn on to review establishments, with the resources currently used to provide periodic review invested instead into quality control for the unit reports. Moving averages could be substituted for intermittent review, with statistical process control methods used to determine when changes in demand sufficient to revise the establishment have occurred.[32]

For some units, variation is such that the estimated establishment may always be imprecisely estimated and our results also highlight that unmeasured influences on demand arising from factors such as unit layout and specialty could have a substantial influence on the staffing requirements. In these cases, the use of professional judgement, already emphasised within the guidance for the SNCT, is paramount. In the face of an apparently objective measure, it is easy to prioritise the measured quantity despite the substantial uncertainty associated with it.[33] It is clear from both these results and the wider literature that professional judgement remains an essential element in determining the required level of nurse staffing.[3 4]

Although our results about other influences on workload were imprecise, both turnover and single rooms have been identified in other research as factors that increase nursing workload,[21–23 34 35] because of the specific work associated with admissions and discharge, increases in indirect care requirements and, for single rooms, the need for increased surveillance of potentially vulnerable patients. The increased workload associated with turnover is acknowledged in the SNCT, where revised multipliers are provided specific to acute admissions units, to reflect the high patient turnover.[18] Our findings could arise because variation in turnover within and between general units is not being sufficiently accommodated within the average demand across all patients. Our finding that nurses on surgical units were less likely to perceive adequate staffing compared with nurses on other (medical or mixed) units could occur if surgical units had a higher workload for a given level of acuity/dependency, which may also result from indirect care associated with surgery, such as arranging transports and providing escorts.[36] This is a novel finding, and where recommendations or mandates for minimum staffing levels exist, medical units are not generally differentiated from surgical units.[37]

While there must be a balance between parsimony and accuracy in any tool, these findings could arise because the current SNCT staffing recommendations fit some units better than others. This lack of fit might be remedied by tailoring the tool and creating versions specific to particular circumstances. Further revisions to the SNCT recommended staffing levels, represented by different 'multipliers' for different unit types beyond the current admissions unit specific multipliers, require further investigation. The wide CIs associated with single rooms and turnover do not directly lend themselves to a formal revision of the multipliers, but the importance of exercising professional judgements about other factors affecting workload is clear.

Although the SNCT multipliers were originally derived using professional expert estimates of time required, subsequent developments have used empirical observations to revise the multipliers.[20] Our study is the first to show that staffing shortfalls, relative to requirement estimated using the tool, are associated with professional judgements that staffing is insufficient to maintain quality care and other indicators that staffing may be inadequate. However, if the SNCT were indicating a level of staffing that was generally judged sufficient to meet all care needs with quality, the relationship between shortfall and staffing adequacy would be expected to diminish as staffing levels increase above the recommended level. Instead the relationships we observed were essentially linear, with no evidence of a threshold above that additional staffing had little effect on the likelihood that nurses would report that there were enough staff. A recent study using the RAFAELA system, widely used in Northern Europe, gave a similar finding. Staffing above the level defined as 'optimal' by the system was associated with decreases in mortality.[38 39] A recent study found that staffing below establishment as determined using the SNCT was associated with an increased risk of death in hospital[11 23] but for registered nurses staffing the relationship was linear, with no threshold. So while our findings are consistent with the SNCT providing a measure of demand, there is no evidence to support the assertion that the recommended staffing levels are optimal in any meaningful sense.

The effects of registered nurse and nurse assistant shortfalls on perceptions of staffing adequacy were similar but independent. Although such a finding might be interpreted as indicating that there is substitution between registered nurses and assistants, the contribution of the two groups to quality and safety is not equivalent they are not interchangeable. A large body of research points to the specific importance of maintaining a rich registered nurse skill mix for patient safety.[40] More recent studies have shown the important contribution of both registered nurses and assistants in maintaining both patient safety and the quality of interpersonal care.[11 41 42] Simple substitutions are not feasible because the contributions of each group are distinct. Effective deployment of assistants is contingent on having sufficient RNs to supervise and support them.[11 41]

The SNCT, while widely used in England, is by no means the only staffing tool available. Given the vast numbers of reports and different tools, it is hard to say definitively that there are no data that would allow comparison of the precision of the SNCT with other tools for estimating staffing establishments, but our reviews found no recent studies giving similar data about other tools.[3 4]

## Limitations

Training was provided to unit nurses in using the SNCT, but the extensive nature of the study is such that the reliability of the ratings we used is likely to be less than that

achieved by expert raters in a dedicated review of establishments. However, the wide variation in precision of estimates of unit establishments is unlikely to be explained by this factor alone. Furthermore, the circumstances of our study resemble routine use of the system when, as is now becoming common, assessments are completed daily by shift leaders. We did find evidence of systematic coding errors for the assessment of staffing adequacy on some units, although our substantive conclusions were unaffected. However, this may indicate that less systematic errors were also occurring. The effect of such errors would be to attenuate our ability to estimate relationships and so we may be underestimating the relationship between staffing shortfalls and perceptions of staffing adequacy. We investigated correlations and causality cannot be assumed. Our study did not explore the relationship between staffing and objective care outcomes; instead our measures of staffing adequacy relied on subjective reports by nurses. However, these subjective assessments have been shown to be associated with important patient outcomes.[43–45] We did not ask staff to report consequences of overstaffing. In our study, the judgement of staffing adequacy and ratings of the SNCT were performed by the same people, so it is possible there is some degree of bias. However, the nature of the SNCT reports—numbers of patients in each category—is sufficiently distinct from the staffing adequacy questions that our findings are unlikely to be a simple product of common method bias. Our sample was large but arose from only four hospitals so we cannot be sure that these results would generalise across all hospitals.

## CONCLUSIONS

In this study, we have asked (and answered) a number of questions about the SNCT. These questions, the precision with which establishments are estimated, the extent to which the averages provided from a patient classification can accommodate variation from factors that are unrelated to individual patients and whether the identified staffing level is in any sense 'optimal', all need to be asked of other tools. Our recent reviews suggest such questions are rarely asked of other tools and are even more rarely answered.

The SNCT can provide a reliable estimate of a unit staffing establishment, but larger samples than the currently recommended minimum are required for most units to provide estimates that are within one whole time equivalent staff member of the mean. For some units, such precision is hard to obtain, and there may be systematic variations in staffing requirements associated with some unit types that are not accounted for by the SNCT. While we recommend further exploration of the factors affecting the reliability and validity of the SNCT estimates and suggest that moving averages instead of periodic reassessments could be used to identify when changes in establishments are needed, our findings also firmly underpin the conclusion that measurement is an adjunct to professional judgements, not a replacement for it. The SNCT does appear to provide a measure of nursing workload, but the recommended staffing levels derived from it are not necessarily optimal.

**Acknowledgements** The authors of this paper wish to acknowledge the contributions of Rosemary Chable, Andrew Dimech, Jeremy Jones, Yvonne Jeffrey, Antonello Maruotti, Alejandra Recio Saucedo and Nicola Sinden, who contributed to the acquisition of funding and the design of the project, and Clare Aspden, Tracey Cassar and Shirley Hunter who contributed to data collection. We also wish to acknowledge all the nurses who answered staffing adequacy questions and completed Safer Nursing Care Tool ratings.

**Contributors** PG (professor, health services research): principal investigator, original conceptualisation of the study and study design, secured funding, oversaw acquisition of the data, data analysis, interpretation of results and drafted article. CS (research fellow, operational research): undertook descriptive and regression analyses and contributed to critical revision of the article. JB (professor, health services research): contributed to study design, acquisition of funding, interpretation of results and critical revision of the article. DC (senior medical statistician): provided statistical advice, contributed to interpretation of results and critical revision of the article. NP (clinical professor, nursing): contributed to study design, acquisition of funding, acquisition of the data, interpretation of results and critical revision of the article. TM (principal research fellow, operational research and data science) contributed to study design, acquisition of funding, oversaw acquisition of the data, data analysis, interpretation of results and critical revision of the article.

**Funding** This report presents independent research funded by the UK's National Institute for Health Research (NIHR) Health Services and Delivery Research Programme number 14/194/21. JB, NP, PG and TM were award holders.

**Disclaimer** The funders had no role in study design, data collection and analysis, decision to publish, or preparation of the manuscript. The views and opinions expressed by authors in this publication are those of the authors and do not necessarily reflect those of the NHS, the NIHR, NETSCC, the PHR programme or the Department of Health and Social Care.

**Competing interests** PG is a member of the National Health Service Improvement safe staffing faculty steering group. The safe staffing faculty programme is intended to ensure that knowledge of the Safer Nursing Care Tool (SNCT), its development and its operational application is consistently applied across the NHS.

**Patient and public involvement** Patients and/or the public were involved in the design, or conduct, or reporting, or dissemination plans of this research. Refer to the Methods section for further details.

**Patient consent for publication** Not required.

**Ethics approval** The study was approved by the Health Research Authority through the integrated research application system (IRAS) approval number 190 548. Ethical approval was granted by the University of Southampton Ethics committee (reference 18809).

**Provenance and peer review** Not commissioned; externally peer reviewed.

**Data availability statement** Data are available in a public, open access repository. All data supporting this article are openly available from the University of Southampton repository at https://doi.org/10.5258/SOTON/D1134.

**ORCID iDs**
Peter Griffiths http://orcid.org/0000-0003-2439-2857
Christina Saville http://orcid.org/0000-0001-7718-5689

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
