## [Reviewer comments · BMJ Open]

ARTICLE DETAILS

TITLE (PROVISIONAL)	Performance of the Safer Nursing Care Tool to measure nurse staffing requirements in acute hospitals: a multi-centre observational study
AUTHORS	Griffiths, Peter; Saville, Christina; Ball, Jane; Culliford, David; Pattison, Natalie; Monks, Thomas

VERSION 1 – REVIEW

REVIEWER	Alison Leary London South Bank University, 412928
REVIEW RETURNED	13-Dec-2019

GENERAL COMMENTS	Thank you for this interesting paper. Comments as follows. Abstract. The objective (isn't there normally an aim?) says the study is to "explore precision" of the SNCT. I wonder if precision is the right word? As we do not know what is precise-is it sensitivity? The design is very brief and needs more detail. The abstract is clear but the result again speak about precision of 4.1% this could do with some clarity-or at least define precision (as its an observational study). Odds are introduced for the first time. The introduction and background are clear the comment about studies not demonstrating tipping points is interesting-we published two papers on this mining routinely collected data and the non linear relationships (2017 and 2019) and so have others. If its a specific issue to SNCT it might be better to say this. Methods are clear but there details on extraction and governance seems to be missing. Ethical approval from the university was obtained but how was trust data obtained? The removal of the uplift is unclear and it would be good to know the rational for this. I also wonder about the integrity of the multipliers in this kind of approach and it is good to this is questioned. My more general comments are around the accessibility of the work. I enjoyed reading this paper but it requires technical knowledge not always accessible to, for example, directors of nursing or COOs. A clearer abstract would help with this.
--

REVIEWER	Annette Richardson The Newcastle upon Tyne Hospitals NHS Foundation Trust I am a co-investigator on a current critical care nurse staffing study and the primary author is also a co-investigator on this study.
REVIEW RETURNED	13-Jan-2020

GENERAL COMMENTS	Thank you for asking me to review the paper. It is extremely well written and uncovers important nurse staffing level knowledge. I only have a couple of points to suggest.
---

	1. Abstract (Results) - sentence starting 'Every registered nurse...' is very long and difficult to comprehend. It would be good to review and break up so clearer. 2. I'm not clear why the focus of the 'micro survey' only concentrated on perceived consequences of lack of nurses. Please could you explain why questions probing over staffing weren't explored or is this a limitation?
--	--

REVIEWER	Ricarda Milstein Universität Hamburg/Hamburg Center for Health Economics Germany
REVIEW RETURNED	27-Jan-2020

GENERAL COMMENTS	Dear Peter, Christina, Jane, David, Natalie and Thomas, dear editors, Thank you very much for submitting this great paper titled "Performance of the Safer Nursing Care Tool to measure nurse staffing requirements in acute hospitals: a multi-center observational study" to BMJ Open. It has been a treat to read it and I strongly welcome this study. As you might have noticed, I am German and will hence assume the position of a non-English/ continental European and comment on selected aspects, which might be difficult to understand for somebody, who is less familiar with the English system. I will largely follow the chronological order of the paper. First, I was wondering whether the SNCT is used as an internal guideline to calculate staffing requirements (This is largely how I understand the NICE guideline?), or whether it has any relevance beyond that, for example for billing purposes. You touch this briefly on page 6, but I'm still not sure whether this triggers a higher DRG with a higher weight, or not. I'd be grateful if you could clarify this. Second, you mention several studies, which have demonstrated an association between nurse staffing levels and quality of care. Here, I would like to encourage you to diversify your sources. I have noticed that you have largely referenced your own work, which I love. However, this might lead to some criticism. It might also be perceived in a critical way by the medical community because your work has largely been published in Nursing Journals. I'm thinking of the many papers by Jack Needleman, Linda Aiken and Joanne Spetz, among others. As they have published in medical journal or health economic journals, such as NJEM, JAMA, Medical Care and HSR, this might be a good addition for the medical community. As your study results also complement this bulk of U.S.-based work, I feel that you could strengthen your argument even further. Third, you mention California as the most notable example for nurse staffing. Please allow me to add Germany, here. Germany has introduced minimum nurse staffing ratios in four department types last year and they have been expanded to a total of six unit types with more unit types being planned. It includes a severe sanctioning mechanism. It requires hospitals to meet minimum staffing ratios in their units with different levels by day and night shift and unit type.
--

Fourth, I would be grateful to learn more about the SNCT. I could learn over the course of this paper that it was developed by professional groups. However, I still do not know how exactly hospitals arrive at their demand for staff. Do nurses assess/classify the nursing requirement of their patients on a daily basis? Or only once at the patient's admission? To me, this is highly relevant to understand the administrative burden of the instrument. If we talk about a daily reassessment of patients (we used to have that as a mandatory instrument in Germany from 1992-1998), this poses a significant burden on nurses and it makes me wonder whether it's worth it.

Fifth, let me talk about the elephant in the room. To me, you do not measure whether NHPPD appropriately capture staffing requirements, but whether it reflects nurses' subjective understanding of being sufficiently staffed. The three questions on page 10 are rather broad. I'd call into question whether not missing a break translates into being adequately staffed. You now measure whether NHPPD correlate with adequacy as surveyed. However, based on that, I have no idea to which extent this translates into better patient safety, such as reduced rates of pressure ulcer, mortality, urinary tract infection, hospital-acquired pneumonia etc. It depends on your outcome, of course. If you want to improve job satisfaction, this work is sufficient. If we move towards patient outcomes, however, I have no idea to which degree meeting staffing adequacy translates into it. Thus, I'd like to encourage you to strengthen this link.

Sixth, on page 20, you argue that surgical units have a higher workload and introduce that as a novel finding. With all due respect, this took me by surprise, because I do not see that being the case. California and Germany vary their ratios based on the unit.

Papers, which address this, are, for example, Sales et al. (2008): The association between nursing factors and patient mortality in the Veterans Health Administration: the view from the nursing unit level. *Med Care* 46(9):938-45. Mark, BA et al. (2004): A longitudinal examination of Hospital Registered Nurse Staffing and Quality of Care. *Health Serv Res* 39(2):279-300. Blegen MA, et al. (2011). Nurse staffing effects on patient outcomes: Safety-net and non-safety-net hospitals. *Med Care* 49(4):406-14.

Finally, I am curious about the absence of a few limitations that I would have expected, here. To begin with, there is the omnipresent question of endogeneity. Next, please make clear that you only investigate correlations, and not causalities. Furthermore, as mentioned earlier, we have no idea to which extent nursing adequacy questions capture "real" adequacy (however defined! I am fully aware of the difficulty of this criticism, but it is still there). This paper does not establish correlations to outcome indicators to substantiate this. None of these limitations do, in any way, diminish the relevance of this paper, but I would recommend to add them here to prevent potential criticism.

I hope that you found my comments helpful. Thank you very much for your attention!

VERSION 1 – AUTHOR RESPONSE

Reviewer(s)' Comments to Author:

Reviewer: 1

Reviewer Name: Alison Leary

Institution and Country: LSBU & USN

Please state any competing interests or state 'None declared': None declared

Please leave your comments for the authors below Thank you for this interesting paper.

Comments as follows.

Abstract. The objective (isn't there normally an aim?) says the study is to "explore precision" of the SNCT. I wonder if precision is the right word? As we do not know what is precise-is it sensitivity?

We are following the journal convention re statement of objectives. We mean precision as in "accuracy" (we define this in the paper- see below). More precisely: How close are estimates of average staffing required to the true average for different sample sizes? We have slightly rephrased and added "for different sample sizes" (pg 2, line 27-28).

The design is very brief and needs more detail.

We have added more detail "...of staffing requirements (estimated using a tool) to professional judgements of adequate staffing. Multi-level logistic regression modelling." (pg 2, line 31-33)

The abstract is clear but the result again speak about precision of 4.1% this could do with some clarity-or at least define precision (as its an observational study).

"defined as half the width of the confidence interval expressed as a percentage of the mean" (pg 2-3, line 44-45)

Odds are introduced for the first time. The introduction and background are clear the comment about studies not demonstrating tipping points is interesting-we published two papers on this mining routinely collected data and the non linear relationships (2017 and 2019) and so have others. If its a specific issue to SNCT it might be better to say this.

Thanks for pointing out these recent papers on non-linear relationships. The discussion is a somewhat nuanced one in that few studies have addressed the issue at all and of those that have some have found one, some have not. It is by no means specific to the SNCT. We have now rephrased somewhat to emphasise the limited evidence as opposed to an implied conclusion and cite the most relevant of your studies to this context: "and few studies have explored tipping points in relationships, which could be one indication of an optimal staffing level.^{2 13 14}" (pg 6, line 115-116)

Methods are clear but there details on extraction and governance seems to be missing. Ethical approval from the university was obtained but how was trust data obtained?

*We obtained R&D approvals from each Trust. We have added revised to add details “The study was approved by the Health Research Authority through the integrated research application system (IRAS) approval number 190548. Ethical approval was granted by the University of Southampton Ethics committee (reference 18809). The study was prospectively registered (ISRCTN 12307968).30 This study did not require NHS Research Ethics Committee approval because no data were collected directly from patients, and all patient data were pseudo-anonymised at source with no sensitive patient data transferred.
(pg 14, line 329-335).*

The removal of the uplift is unclear and it would be good to know the rational for this.

We have added “The SNCT calculation gives the number of staff to employ (staffing establishment) including an uplift for staff leave and an allowance for sickness, so we converted this into the implied daily hours of staff time available.” (pg 10, line 214-216) and “We assumed that this uplift used in the tool was enough for the staff employed to be able to cover all long-term absences.” (pg 10, line 218-219).

I also wonder about the integrity of the multipliers in this kind of approach and it is good to this is questioned.

My more general comments are around the accessibility of the work. I enjoyed reading this paper but it requires technical knowledge not always accessible to, for example, directors of nursing or COOs. A clearer abstract would help with this.

Thank you. We hope that our rewrite of the abstract based on comments above has helped with this.

Reviewer: 2

Reviewer Name: Annette Richardson

Institution and Country: The Newcastle upon Tyne Hospitals NHS Foundation Trust

Please state any competing interests or state ‘None declared’:

I am a co-investigator on a current critical care nurse staffing study and the primary author is also a co-investigator on this study.

Please leave your comments for the authors below Thank you for asking me to review the paper. It is extremely well written and uncovers important nurse staffing level knowledge. I only have a couple of points to suggest.

Thank you.

1. Abstract (Results) - sentence starting ‘Every registered nurse...’ is very long and difficult to comprehend. It would be good to review and break up so clearer.

We have broken up this sentence and rewritten as recommended:

“When staffing was lower than that required according to the SNCT, for each hour per patient day of registered nurse staffing below the required staffing level, the odds of nurses reporting that there were enough staff to provide quality care were reduced by 11%. Correspondingly, the odds

of nurses reporting that necessary nursing care was left undone were increased by 14%.” (page 3, line 47-52).

2. I'm not clear why the focus of the 'micro survey' only concentrated on perceived consequences of lack of nurses. Please could you explain why questions probing over staffing weren't explored or is this a limitation?

This is a limitation. When designing the survey, we used questions from the large European “RN4Cast” study, which does not include any questions about overstaffing. We have added “We did not ask staff to report consequences of overstaffing.” (pg 24, line 533-534)

Reviewer: 3

Reviewer Name: Ricarda Milstein

Institution and Country:

Universität Hamburg/Hamburg Center for Health Economics Germany Please state any competing interests or state 'None declared': None declared

Please leave your comments for the authors below Dear Peter, Christina, Jane, David, Natalie and Thomas, dear editors,

Thank you very much for submitting this great paper titled “Performance of the Safer Nursing Care Tool to measure nurse staffing requirements in acute hospitals: a multi-center observational study” to BMJ Open. It has been a treat to read it and I strongly welcome this study.

Thank you.

As you might have noticed, I am German and will hence assume the position of a non-English/continental European and comment on selected aspects, which might be difficult to understand for somebody, who is less familiar with the English system. I will largely follow the chronological order of the paper.

First, I was wondering whether the SNCT is used as an internal guideline to calculate staffing requirements (This is largely how I understand the NICE guideline?), or whether it has any relevance beyond that, for example for billing purposes. You touch this briefly on page 6, but I'm still not sure whether this triggers a higher DRG with a higher weight, or not. I'd be grateful if you could clarify this.

Yes that's right, it is for hospitals to use as an internal guideline. No, it doesn't affect billing, only costs to the hospital. We have added “It is not however used for billing purposes since in England billing is based on activity and does not explicitly account for nursing staff.”(pg 7, line 143-145).

Second, you mention several studies, which have demonstrated an association between nurse staffing levels and quality of care. Here, I would like to encourage you to diversify your sources. I have noticed that you have largely referenced your own work, which I love. However, this might lead to some criticism. It might also be perceived in a critical way by the medical community because your work has largely been published in Nursing Journals. I'm thinking of the many papers by Jack Needleman, Linda Aiken and Joanne Spetz, among others. As they have published in medical journal

or health economic journals, such as NJEM, JAMA, Medical Care and HSR, this might be a good addition for the medical community. As your study results also complement this bulk of U.S.-based work, I feel that you could strengthen your argument even further.

We do understand the point you are making here but the reason for our choices is the large body of literature demonstrating associations between nurse staffing levels and quality of care, therefore we concentrated on citing recent substantial reviews (2, 7-9) of which two are from US authors in US medical journals, and examples of recent studies (10-12), some of which are indeed our own papers. Most of the papers cited are in "medical journals" e.g. Med Care, Lancet, Ann Intern Med, BMJ Quality and Safety. Reference 10 is a paper by Aiken et al.

Third, you mention California as the most notable example for nurse staffing. Please allow me to add Germany, here. Germany has introduced minimum nurse staffing ratios in four department types last year and they have been expanded to a total of six unit types with more unit types being planned. It includes a severe sanctioning mechanism. It requires hospitals to meet minimum staffing ratios in their units with different levels by day and night shift and unit type.

Thank you, that is useful to know and to illustrate the global scope of these policies. We have amended to "for example in California in the USA, some Australian states and more recently in Germany."¹³

Fourth, I would be grateful to learn more about the SNCT. I could learn over the course of this paper that it was developed by professional groups. However, I still do not know how exactly hospitals arrive at their demand for staff. Do nurses assess/classify the nursing requirement of their patients on a daily basis? Or only once at the patient's admission? To me, this is highly relevant to understand the administrative burden of the instrument. If we talk about a daily reassessment of patients (we used to have that as a mandatory instrument in Germany from 1992-1998), this poses a significant burden on nurses and it makes me wonder whether it's worth it.

Yes it is done at least once per day, so quite a high admin requirement. We have added "At least once per day, patients occupying beds on the ward..." (pg 7, line 145-146).

Fifth, let me talk about the elephant in the room. To me, you do not measure whether NHPPD appropriately capture staffing requirements, but whether it reflects nurses' subjective understanding of being sufficiently staffed. The three questions on page 10 are rather broad. I'd call into question whether not missing a break translates into being adequately staffed. You now measure whether NHPPD correlate with adequacy as surveyed. However, based on that, I have no idea to which extent this translates into better patient safety, such as reduced rates of pressure ulcer, mortality, urinary tract infection, hospital-acquired pneumonia etc. It depends on your outcome, of course. If you want to improve job satisfaction, this work is sufficient. If we move towards patient outcomes, however, I have no idea to which degree meeting staffing adequacy translates into it. Thus, I'd like to encourage you to strengthen this link.

This is of course an important issue and we are only making a start here. However, professional judgement of staffing adequacy should not be dismissed. We have added "Therefore we used professional judgement as the "gold standard", as we found no evidence that any tool provides a more accurate measure of the staffing required." (pg 7-8, line 160-165). We also explained in the Limitations that "Our study did not explore the relationship between staffing and objective care outcomes; instead our measures of staffing adequacy relied on subjective reports by nurses. However, these subjective assessments have been shown to be associated with important patient outcomes⁴³⁻⁴⁵." (pg 25, line 544-546)

We have added "We chose to use three items for pragmatic reasons, since we judged that more would have been too much of an administrative burden and might have resulted in poorer data

quality.” (pg 11, line 252-254) These questions represent different indicators of having enough staff. We agree with you that taking a break does not imply adequate staffing, however we were interested to see whether higher rates of missed breaks corresponded with larger staffing shortfalls according to the tool.

Agreed, we did not explicitly study the link to patient outcomes here. One of the limitations is “The study did not explore the impact on objective care outcomes.” (pg 4, line 81 & pg 25 line 558) although we also note “A recent study found that staffing below establishment as determined using the SNCT was associated with an increased risk of death in hospital ^{11, 23} (p23 lines 521-523)”

Sixth, on page 20, you argue that surgical units have a higher workload and introduce that as a novel finding. With all due respect, this took me by surprise, because I do not see that being the case. California and Germany vary their ratios based on the unit.

Papers, which address this, are, for example, Sales et al. (2008): The association between nursing factors and patient mortality in the Veterans Health Administration: the view from the nursing unit level. *Med Care* 46(9):938-45. Mark, BA et al. (2004): A longitudinal examination of Hospital Registered Nurse Staffing and Quality of Care. *Health Serv Res* 39(2):279-300. Blegen MA, et al. (2011). Nurse staffing effects on patient outcomes: Safety-net and non-safety-net hospitals. *Med Care* 49(4):406–14.

We have clarified the finding that is novel by writing “Our finding that nurses on surgical units were less likely to perceive adequate staffing compared to nurses on other (medical or mixed) units” (p22 line 492-494). We perhaps phrased the discussion incorrectly as we are not asserting that surgical units have a higher workload—merely that this could explain the findings. We have rephrased this “Our finding ... could occur if surgical units had a higher workload for a given level of acuity / dependency” (p22 line 492-494) However we are familiar with some of the sources you cite and have looked at the others and we do not think they change the point we are making (please forgive us if we have missed anything). Regarding our assertion that mandatory staffing policies do not differentiate between medical and surgical units - for example in California the ratio of 1:5 is given for med/surg units and the legislation in Victoria gives ratios for acute medical surgical wards by hospital type – there is no variation for medical or surgical.

Finally, I am curious about the absence of a few limitations that I would have expected, here. To begin this, there is the omnipresent question of endogeneity. Next, please make clear that you only investigate correlations, and not causalities. Furthermore, as mentioned earlier, we have no idea to which extend nursing adequacy questions capture “real” adequacy (however defined! I am fully aware of the difficulty of this criticism, but it is still there). This paper does not establish correlations to outcome indicators to substantiate this. None of these limitations do, in any way, diminish the relevance of this paper, but I would recommend to add them here to prevent potential criticism.

Regarding endogeneity, we have added “In our study the judgement of staffing adequacy and ratings of the SNCT were performed by the same people, so it is possible there is some degree of bias. However, the nature of the SNCT reports – numbers of patients in each category – is sufficiently distinct from the staffing adequacy questions that our findings are unlikely to be a simple product of common method bias.” (pg 25, line 549-554)

We have added “We investigated correlations and causality cannot be assumed.” (pg 25, line 544-545)

We have added “Our study did not explore the relationship between staffing and objective care outcomes; instead our measures of staffing adequacy relied on subjective reports by nurses.” (pg 25, line 545-546)

I hope that you found my comments helpful. Thank you very much for your attention!

Thanks very much for your detailed comments which were very helpful.

VERSION 2 – REVIEW

REVIEWER	Alison Leary London South Bank University, 412928
REVIEW RETURNED	09-Mar-2020

GENERAL COMMENTS	Thank you I look forward to the publication.
--

REVIEWER	Annette Richardson The Newcastle upon Tyne Hospitals NHS Foundation Trust I am a co-investigator on a current NIHR funded critical care nurse staffing study and the primary author is also a co-investigator on this study.
REVIEW RETURNED	11-Mar-2020

GENERAL COMMENTS	Thank you for making the changes
----------------------------------

REVIEWER	Ricarda Milstein Universität Hamburg/Hamburg Center for Health Economics Germany
REVIEW RETURNED	25-Mar-2020

GENERAL COMMENTS	Thank you very much for submitting a revised version of your manuscript entitled “Performance of the Safer Nursing Care Tool to measure nurse staffing requirements in acute hospitals: a multi-centre observational study. First, to get this off my chest, I would like to apologise for comment No. 2 in my previous (first) review of your manuscript. I had suggested that you add more sources and recommended Jack Needleman, Linda Aiken and Joanne Spetz as examples. Only after submitting my comments, I had to note that they were already all there, and that I must have gotten lost in your references. Hence, I would like to apologise for that comment, which must have taken you by surprise. Second, thank you very much for your detailed replies to my comments. I welcome your clarifications. For example, your reply to point sixth of my previous comments has helped me to better understand the difference of this paper to already existing ones, and to appreciate its novelty much better. I remain a bit concerned about the administrative burden arising from this instrument, but this stems largely from my own country’s experience from daily written patient assessments and from discussions on an ever-increasing burden on the health workforce. There are a few questions that this manuscript opens up, but which are beyond the scope of this paper. For example, I am thinking along the lines of the interaction between subjective and objective staffing measures and patient outcomes. This, however, is for another paper. To sum this up, the authors (and mostly Christina, I suppose?), have responded comprehensively and sufficiently to my
--

	comments. Thank you very much for your answers and for your submission to BMJ Open.
--	---